# HIF-1α Negatively Regulates Irisin Expression Which Involves in Muscle Atrophy Induced by Hypoxia

**DOI:** 10.3390/ijms23020887

**Published:** 2022-01-14

**Authors:** Shiqiang Liu, Pengyu Fu, Kaiting Ning, Rui Wang, Baoqiang Yang, Jiahui Chen, Huiyun Xu

**Affiliations:** 1Key Laboratory for Space Bioscience and Biotechnology, School of Life Sciences, Northwestern Polytechnical University, Xi’an 710072, China; liushiqiang199204@mail.nwpu.edu.cn (S.L.); ningkaiting@mail.nwpu.edu.cn (K.N.); 2020204576@mail.nwpu.edu.cn (R.W.); yangbq@mail.nwpu.edu.cn (B.Y.); chenjiahui@mail.nwpu.edu.cn (J.C.); 2Department of Physical Education, Northwestern Polytechnical University, Xi’an 710072, China; fupy@nwpu.edu.cn; 3Research Center of Special Environmental Biomechanics & Medical Engineering, Northwestern Polytechnical University, Youyi Xilu 127, Xi’an 710072, China

**Keywords:** hypoxia, muscle atrophy, HIF-1α, irisin, FNDC5

## Abstract

Exposure to high altitude environment leads to skeletal muscle atrophy. As a hormone secreted by skeletal muscles after exercise, irisin contributes to promoting muscle regeneration and ameliorating skeletal muscle atrophy, but its role in hypoxia-induced skeletal muscle atrophy is still unclear. Our results showed that 4 w of hypoxia exposure significantly reduced body weight and gastrocnemius muscle mass of mice, as well as grip strength and the duration time of treadmill exercise. Hypoxic treatment increased HIF-1α expression and decreased both the circulation level of irisin and its precursor protein FNDC5 expression in skeletal muscle. In in vitro, CoCl_2_-induced chemical hypoxia and 1% O_2_ ambient hypoxia both reduced FNDC5, along with the increase in HIF-1α. Moreover, the decline in the area and diameter of myotubes caused by hypoxia were rescued by inhibiting HIF-1α via YC-1. Collectively, our research indicated that FNDC5/irisin was negatively regulated by HIF-1α and could participate in the regulation of muscle atrophy caused by hypoxia.

## 1. Introduction

Skeletal muscles account for more than 40% of the body and are responsible for converting chemical energy into mechanical energy to generate force and power, maintain posture, and produce movement [1]. Long-term exposure to a high-altitude environment can cause skeletal muscle atrophy, especially above 5000 m [2] for the first time [3]. Although studies have reported that ambient hypoxia in the high-altitude region can be the main reason for skeletal muscle atrophy, the mechanism is still unclear.

Previous studies have shown that basic helix-loop-helix (bHLH) transcription factors—myogenic differentiation 1 (Myod1), myogenic factor 5 (Myf5), myogenin (MyoG), and myogenic regulatory factor 4 (Mrf4)—are important for skeletal muscle hyperplasia and hypertrophy [4]. Additionally, as the central regulator in sensing and responding to hypoxia, hypoxia-induced factor-1 α (HIF-1α) plays a vital role in regulating skeletal muscle atrophy via controlling the expression of MyoG and myosin heavy chain type II (MyHc) [5,6] and the distribution of skeletal muscle fibers [7] as well as the formation of multinucleated myotubes [5,8].

The expression of fibronectin type III domain-containing protein 5 (FNDC5) is regulated by the peroxisome proliferator-activated receptor-γ coactivator-1α (PGC-1α) [9] and myostatin (Mstn) [10,11] in muscles [9]. In the stimulation of exercise, irisin was cleaved at the extracellular domain of FNDC5 by splicing enzyme A Disintegrin and Metalloproteinase Domain 10 (ADAM10) [12]. Irisin was regarded as a biomarker of sarcopenia [13] and highly related to muscle atrophy [14]. Additionally, injecting recombinant irisin (r-irisin) into skeletal muscles rescued denervation [15] or hindlimb unloading [16] induced muscle atrophy. In vitro, the expression trend of FNDC5/irisin was consistent with MyoG during human myotubes differentiation [17]. Previous studies have also found that hypoxia inhibits irisin expression in cardiomyocytes [18] and serum of volunteers who participated in the Alps climbing [19]. Nevertheless, the potential mechanism is still not clear. 

In this study, we investigated the role of FNDC5/irisin in muscle atrophy caused by hypoxia in vivo and in vitro and identified that the decreased expression of FNDC5/irisin and the myogenic factors *Mrf4* and *MyoG* may be important reasons in hypoxia-induced muscle atrophy. We also found that the expression of FNDC5/irisin was negatively regulated by HIF-1α in hypoxia. 

## 2. Results

### 2.1. Hypoxia Reduced the Lean Weight and Fat Weight of Mice 

The body composition of mice was detected by Dual Energy X-ray Absorptiometry (DEXA) after 2 or 4 w of hypoxic exposure (Figure 1A). Body weight (Figure 1B), lean weight (Figure 1C), and fat weight (Figure 1D) were significantly reduced but not fat in tissue (Figure 1E). However, hypoxia had a slight effect on bone-related parameters (Figure 1F–I); only bone mineral content (BMC) and bone volume were decreased by 2 w of hypoxic treatment but recovered to the control level at the end of 4 weeks of hypoxia (Figure 1H,I). In general, hypoxia significantly reduced lean weight and fat weight, which could be the main reason for body weight loss.

### 2.2. Four weeks of Hypoxic Exposure Induced Muscle Atrophy of Mice

The skeletal muscle function of mice was judged by the grip strength and the duration time of exercise. The grip strength (Figure 2A) was significantly reduced at the end of 2 and 4 w of hypoxia (*p* < 0.01). Additionally, a significant reduction of 59.6% (*p* < 0.01) in the duration time of treadmill exercise was observed in mice exposed to 28 days of hypoxia (Figure 2B). In addition, the relative weight of skeletal muscles (gastrocnemius, soleus, quadriceps, and triceps) was normalized to their tibia lengths, and the results showed that gastrocnemius (Gas) and quadriceps (Quad) muscle weight were reduced after 4 w of hypoxic exposure, while the soleus (Sol) and triceps (Tric) were not changed (Figure 2C). Moreover, the muscle fibers were stained by H&E. The cross-sectional area (CSA) and the cross-sectional diameter of the Gas muscles were significantly reduced after hypoxic treatment (Figure 2D–F), and hypoxic treatment resulted in a decrease in the number of muscle fibers larger than 1500 μm^2^ in area and larger than 40 μm in diameter (Figure 2G–H). Moreover, Sol muscles were not changed (Appendix A). The expression of myogenic factors was detected by q-PCR in the Gas muscle of mice. The results showed that expression of *Mrf4* and *MyoG* were significantly reduced, while the expression of *Myf5*, *Myod1*, and *myostatin* (*Mstn*) was not affected by hypoxic treatment (Figure 2I).

### 2.3. Hypoxia Significantly Reduced the Expression of FNDC5/Irisin in Mice

The results of the treadmill experiment showed that 4 w of hypoxic exposure significantly increased the expression of HIF-1α in Gas muscle (Figure 3A,B), meanwhile decreasing the expression of FNDC5 (Figure 3C) in Gas muscle. In addition, irisin expression in plasma was reduced (Figure 3D). However, in the Gas muscle, both *FNDC5* and *ADAM10* (splicing FNDC5 to irisin) mRNA expressions were not changed (Figure 3E,F). Moreover, PGC-1α expression was not affected by hypoxia both in protein and mRNA level (Figure 3G–I). Additionally, myostatin concentration in plasma (Figure 3J) and quadriceps muscle (Figure 3H) was not changed by hypoxic treatment. 

### 2.4. Inhibition of HIF-1α Induced by CoCl_2_ Increased FNDC5 Expression in C2C12 Cells

The relationship between HIF-1α and irisin in hypoxia was investigated. C2C12 myotubes were treated with CoCl_2_ at a concentration of 10, 50, 100, and 200 μM. The cell viability was detected by the cell counting kit 8 (CCK-8). We found that CoCl_2_ did not affect the activity of C2C12 myotubes (Appendix A), even at the concentration of 200 μM. The expression of HIF-1α showed almost a dose-dependent increasing trend with the increase in CoCl_2_ (Figure 4A,B), and correspondingly, the expression of FNDC5 (precursor of irisin) was decreased in a CoCl_2_ concentration-dependent manner (Figure 4C). However, quantitative analysis showed there was no difference in the expression of PGC-1α (Figure 4D). Additionally, FNDC5 protein correlated negatively with HIF-1α (r = −0.627, *p* = 0.0135, Figure 4F), and no statistically significant correlation was observed in the protein expression between FNDC5 and PGC-1α (r = 0.055, *p* = 0.843, Appendix A).

Since HIF-1α is negatively related to the expression of FNDC5 in hypoxia, whether inhibiting HIF-1α could rescue the expression of FNDC5? Here, we used Lificiguat (also named YC-1) as an effective HIF-1α inhibitor and observed the expression of FNDC5. First, through the cell viability test (CCK-8), 1 and 50 μM YC-1 had no effects on cell viability (Appendix A), so 50 μM was chosen for the next experiments [20]. On the 5th day of differentiation, cells were added with 50 or 100 μM CoCl_2_ and 50 μM YC-1 and incubated for another 24 h (Figure 4F). The results showed that both 50 μM and 100 μM CoCl_2_ significantly increased the expression of HIF-1α, and YC-1 reversed the increase only in 100 μM of CoCl_2_ treatment (Figure 4G). In the same way, YC-1 reversed the decrease in FNDC5 expression caused by 100 μM of CoCl_2_ (Figure 4H). However, the expression of PGC-1α did not change during the whole experiment (Figure 4I). 

### 2.5. Inhibition of HIF-1α Induced by 1% O_2_ Ambient Hypoxia Increased FNDC5 Expression and Myotube Formation in C2C12 Cells

After 5 days of differentiation, C2C12 cells were placed in a 1% O_2_ hypoxic chamber with 50 μM YC-1 and cultured for 6 h, 12 h, and 24 h, respectively (Figure 5A). We found that the expression of HIF-1α significantly increased in ambient hypoxia at 12 h and 24 h, which was abrogated by YC-1 treatment (Figure 5B). Additionally, YC-1 treatment rescued the decrease in FNDC5 after 12 h of hypoxia (Figure 5C). The expression of PGC-1α after 12 h of hypoxia was significantly higher than that of 6 h (Figure 5D). 

To investigate whether increasing the expression of irisin in hypoxia can improve hypoxia-induced skeletal muscle atrophy, the C2C12 myotubes were exposed to hypoxia with YC-1 for 12 h, and the diameter and area were detected after the C2C12 myotubes were stained by anti-MyHC (Figure 5E). We found that hypoxia significantly reduced the area (Figure 5F) and diameter (Figure 5G) of C2C12 myotubes, which was reversed by inhibition of HIF-1α by treatment with YC-1. There was no obvious difference observed in the myotube fusion rate (Figure 5H). 

## 3. Discussion

Irisin has been identified as a pro-myogenic factor for ameliorating muscle atrophy, but whether it plays some role in hypoxia-induced muscle atrophy was not investigated. In the present study, we found that irisin expression was negatively regulated by HIF-1α, which may be an important reason for skeleton atrophy caused by hypoxia. 

We found that the weight loss caused by hypoxia is related to the decrease in fat and muscle mass. Here, we mainly focused on the reason for the decrease in muscle mass, which was manifested by decreased grip strength and run duration time of mice, as well as decrease in CSA and diameter of muscle fiber. To date, the cellular mechanism of hypoxia-induced skeletal muscle atrophy has been concerned with the disorder of protein synthesis and degradation [21,22,23] and hormone secretion [24] in skeletal muscle, etc. However, the mechanism is still unclear. 

In this study, the decrease in muscle mass was only observed in gastrocnemius muscle, not in soleus muscle. We thought that might be due to the difference in fiber composition of gastrocnemius and soleus. The gastrocnemius muscle was mainly composed of type Ⅱ muscle fiber, while soleus muscle is mainly composed of type Ⅰ muscle fiber, which was thought more insensitive than type II fiber to hypoxia in a previous study [25]. The level of circulating irisin has been regarded as a newly identified biomarker for muscle weakness and atrophy [26]. The concentration of irisin was lowered in sarcopenia and pre-sarcopenia groups compared with non-sarcopenic participants [13,27], but ascent when these participants increased skeletal muscle mass or enhanced skeletal muscle function through high [28] or low [29] intensity exercise. Reza MM et al. have found that irisin increases myogenic differentiation and myoblast fusion by activating IL-6 signaling and further improves injured skeletal muscle by promoting protein synthesis [15]. Moreover, Chang JS et al. have revealed that irisin prevents dexamethasone-induced atrophy in C2C12 myotubes [30]. Irisin came from FNDC5 and was cleaved by splicing enzyme ADAM10 [12]. ADAM10 was reported to be modulated by HIF-1α [31,32], which could be induced by hypoxia. Then for hypoxia-induced skeletal muscle atrophy, does irisin play a role?

Our results showed that, along with skeletal muscle atrophy caused by hypoxia, both irisin in plasma and FNDC5 in gastrocnemius muscle were reduced. Additionally, irisin precursor FNDC5 was reduced in C2C12 myotubes both by ambient hypoxia and CoCl_2_-induced hypoxia in vitro. This is consistent with the discovery of decreased irisin in humans after 2 w of climbing on the Alps [19]. Moscoso I et al. showed that FNDC5/irisin expression was also lowered in H9C2 cardiomyocytes by 0.1% O_2_ hypoxia [18]. Thus, the expression of FNDC5/irisin probably is a biomarker in hypoxia-induced muscle atrophy. Moreover, we found that the reduction in FNDC5/irisin expression induced by hypoxia can be reversed by inhibition of HIF-1α, which was confirmed by that YC-1 rescued the diameter and area of myotubes in 12 h of hypoxia in vitro. We also found that 6 h of hypoxia did not affect the expression of HIF-1α and FNDC5, which could be explained that the time of hypoxic exposure was too little to cause the response. That was in keeping with the result from Moscoso I et al. [18], who have found that FNDC5 expression is decreased in hypoxia at least after 8 h of treatment. Maybe there is a window phase for myotubes to respond to hypoxia [33]. Because 6 h of exposure did not cause the change of HIF-1α and FNDC5, and after 24 h of exposure, the response was also lower than that of 12 h. A future further experiment is required to explain the result.

Myostatin was regarded as the upstream regulator of FNDC5 [9]. Shan et al. have revealed that myostatin stimulates the expression of FNDC5/irisin through the AMPK/PGC-1α pathway [11] in muscle. After that, Ge et al. have found that myostatin post-transcriptionally inhibits Fndc5 expression in both myoblasts and adipocytes via the miR-34a pathway [10]. Conversely, in this study, though FNDC5/irisin expression was decreased by hypoxia, the concentration of myostatin neither in plasma nor quadriceps was changed. Similarly, Sliwicka E et al.’ found that the concentration of myostatin in humans was not changed after 2 w of Alps climbing [19] when irisin was lowered. Additionally, previously the transcriptional coactivator PGC-1α has reported an increase in C2C12 myotubes in severe hypoxic conditions, such as 0.2% [34] or 0.5% O_2_ [35]. Conversely, there were also some opposite results that PGC-1α was reduced by 35% in muscles after 66 d of exposure at an altitude beyond 6400 m [36]. Our results showed that PGC-1α was unchanged by hypoxia compared with normoxia. These competing results showed that PGC-1α might be regulated by many factors, which cannot be seen as a key marker factor in hypoxia-induced muscle atrophy. 

Although these discoveries were revealed in this study, there are still some limitations. The specific mechanism by which HIF-1α regulates irisin expression was not explained. Of note, we only focused on the role of irisin in the initial stage of hypoxia on C2C12 myotubes, and the effect of long-term chronic hypoxia on C2C12 myotubes still requires further exploration.

To summarize, our findings showed that FNDC5/irisin was negatively regulated by HIF-1α and participated in the regulation of muscle atrophy caused by hypoxia. Irisin could be a new challenge and opportunity to treat and improve hypoxia-induced skeletal muscle atrophy in the future.

## 4. Materials and Methods

### 4.1. Mice and Hypoxic Exposure

Sixty male C57Bl/6J mice aged 11–13 weeks were purchased from the Air Force Military Medical University. These mice were randomly divided into two groups: control and hypoxia. In the control group, 30 mice were housed in a normoxic environment with an altitude of 399 m (20.9% O_2_), the rest of the mice were firstly placed in a hypoxic chamber (Guizhou Fenglei Aviation Ordnance Co., Ltd., Guizhou, China, FLYDWC50-ⅡC) with 15.4% (3000 m altitude simulation) for 3 days to adapt the environment, then housed in 11.6% O_2_ (altitude of 5300 m). During the whole experiment, all mice were free to access food and water, and the light and dark cycle was 12 h: 12 h. 

### 4.2. Body Composition and Grip Strength Measurement 

The body composition of mice was detected by a Dual-energy X-ray Absorptiometry machine (DEXA, Medikors Company, Seoul, South Korea, I-BMD) at the day before hypoxic exposure and the end of 2 and 4 weeks of hypoxic exposure. The grip strength of mice was detected by grip force instrument (Jinan Yiyan Technology, Jinan, China, YLS-13A) at the end of 2 and 4 weeks of hypoxic treatment. 

### 4.3. Treadmill Experiment

The mice were subjected to a single treadmill exhaustive exercise to detect the muscle fatigue strength according to a previous protocol [37] at a speed of 14 m/min on a 5° slope by using a Mouse treadmill machine (Anhui Zhenghua Biological Instruments, Anhui, China, ZH-PT) at the end of 4 weeks of hypoxic exposure. 

### 4.4. ELISA for Determining Irisin and Myostatin Concentration in Plasma/Muscle Homogenate

After exercise, the mice were anesthetized by pentobarbital (1.5%, 40 mg/kg), the blood was extracted from the heart and then harvested into EDTA-containing tubes and centrifuged at 3000 rpm for 15 min at 4 ℃, and finally, the plasma was collected [11]. Quadriceps were harvested and then rinsed with pre-cooled PBS at 1:9 weight-volume ratio and then further fully ground with protease inhibitor (Beyotime; P1010; 1:100). Finally, the homogenate was centrifuged at 5000 g for 10 min, and the supernatant was obtained to acquire the quadriceps homogenate. The level of irisin and myostatin in plasma/muscle homogenate was quantified using an ELISA kit according to the protocol of the manufacturer (irisin, Phoenix Pharmaceuticals, Burlingame, CA, USA, NO. EK-067-29/ myostatin, Jianglai, Zhejiang, China, JL50880). 

### 4.5. Muscle Fiber Cross-Sectional Area (CSA) Measurement 

Transverse sections (4 µm) were cut from the mid-belly area of paraffin-embedded gastrocnemius and soleus muscle and then stained with Gill’s hematoxylin followed by 1% eosin (servicebio; H&E; G1005). The slides were scanned using Digital Slice Scanner (KFBIO, Zhejiang, China, KF-PRO-020), CSA of more than 20 muscle fibers from 6 random fields were analyzed (*n* = 6 mice per treatment group) by using ImageJ software.

### 4.6. Cell culture and Hypoxic Treatment 

C2C12 murine myoblasts were purchased from the National Collection of Authenticated Cell Cultures (#GNM 26) and cultured in the growth medium (GM) made of Dulbecco’s modified Eagle’s medium (DMEM; Gibco; 12800-017) with a high glucose concentration supplemented with 10% (*v*/*v*) fetal bovine serum (FBS; BI; C04001-500) and 1% penicillin/streptomycin (Beyotime, Shanghai, China, C0222) at 37 ℃ in a 5% CO_2_ and 95% air-humidified atmosphere. C2C12 myoblasts were plated at the concentration of 7000 cells/cm^2^ [38]. When they grew up to 70–80% confluency, the medium was replaced to differentiation medium (DM) containing DMEM supplemented with 2% (*v*/*v*) horse serum (HS, Gibco, Waltham, MA, USA, 26050088) [38]. Two methods were used to simulate hypoxia in vitro. Chemical hypoxia was induced by CoCl_2_ treatment with a concentration of 10, 50, 100, and 200 μM in DM, while ambient hypoxia was mimicked by being cultured in a cell culture chamber with a gas mixture containing 1% O_2_, 5% CO_2_, and 94% N_2_. 

### 4.7. Cell Viability Assay

The effect of CoCl_2_ (Kemiou, Tianjin, China) or YC-1 (MedChemExpress, Shanghai, China, MCE, HY-14927) treatment on cell viability was detected by cell kit counting-8 (TargetMol, Boston, MA, USA; C0005). Briefly, 2.5 × 10^4^ C2C12 murine myoblasts were seeded in a 96-well plate, after 70–80% confluency, cells were cultured in DM for another 48 h, and then incubated with CoCl_2_ (10, 50, 100, and 200 μM) or YC-1 (1, 5, 10, 20, and 50 μM) and CCK-8 regent (10 μL for each well) for 24 h. The luminescence signal was detected by Multifunction Microplate Reader (Bio-tek, Winooski, VT, USA; Synergy HT).

### 4.8. RNA Extraction and Real-Time Quantitative PCR (qPCR)

Total RNA was isolated from gastrocnemius muscle by using Trizol (AG, Xi’an, China, AG21102), total RNA (1 μg) was reverse-transcribed to cDNA by using HiScript II Q RT SuperMix (Vazymek, Nanjing, China, R223-01). Quantitative real-time polymerase chain reactions (qPCR) were performed in 10 μL final volume system, respectively, 4.6 μL of cDNA template, 0.2 μL primers, and 5 μL 2 × ChamQ SYBR qPCR Master Mix (Vazyme, Nanjing, China, Q311-02). GAPDH was used as the housekeeping gene control. Relative mRNA expression was performed in the comparative dCt method (2^−ΔΔCt^). The primer sequences were synthesized by Sangon Biotech and shown in Table 1.

### 4.9. Western Blotting

Gastrocnemius muscle tissues or C2C12 myotubes were lysed in RIPA enhancer buffer (Beyotime, Shanghai, China, P0013B) and supplemented with protease inhibitor cocktail (Beyotime, Shanghai, China, P1005, 1:100). The lysates/proteins were loaded at 20 μg per lane, after electrophoresis and transfer, the PVDF membrane was incubated with primary antibody, the following primary antibodies: anti-HIF-α (CST, Danvers, MA, USA, #36169, 1:1000), anti-FNDC5 (Abcam, Waltham, MA, USA, ab174833; 1:2000), anti-β-tubulin (Abcam, Waltham, MA, USA, ab6046, 1:2000), anti-GAPDH (Abcam, Waltham, MA, USA, ab8245, 1:1000) and anti-PGC-1α (Proteintech, Wuhan, China20658-1-AP, 1:2000) were incubated overnight at 4 ℃. The HRP-coupled secondary antibody (Immunoway, Jiangsu, China, RS0001, 1:10,000) was incubated for 1 h at room temperature (RT). Protein bands were displayed via clarity^TM^ western ECL substrate (BIO-RAD, Hercules, CA, USA #170-5060) by using Chemiluminescence Apparatus (Tanon, Shanghai, China, T5200). 

### 4.10. Immunofluorescence Staining

C2C12 myotubes were exposed to 1% ambient hypoxia with YC-1 (50 μM) for 12h, and then washed with phosphate saline buffer (1 × PBS) and fixed for 10 min in 4% (*w*/*v*) paraformaldehyde at RT. After being washed 3 times with PBS containing 0.1% Tween 20, myotubes were permeabilized in Triton X-100 (Beyotime, Shanghai, China, P0096) for 10 min at RT and then washed 3 times and incubated with 1% BSA for 30 min. Mouse anti-myosin (Myosin heavy chain, MF-20, DSHB, Iowa City, IA, USA) diluted 1:200 in PBST was added to incubate at 4 ℃ overnight. The cells were washed three times in PBS for 5 min each wash. Alexa Fluor^TM^ 488 goat anti-mouse antibody (Invitrogen, Waltham, MA, USA, R37120) in 1% BSA was added for 1 h a light-proof box at RT. Cell nuclei were stained with Hoechst 33258 (Beyotime, Shanghai, China, C1011) for 10 min. Myotube fusion index (an index reflecting the degree of muscle differentiation, which refers to the total number of nuclei in the myotube with more than 2 nuclei/the total number of nuclei in the same field of vision), myotube diameter (Feret’s diameter) and area were measured by ImageJ software. 

### 4.11. Statistical Analyses

All assays were performed at least 3 replicates, and the quantitative experimental data were presented as mean ± standard deviation (SD). The statistical analysis was performed with GraphPad Prism 8.0 (GraphPad Software, San Diego, CA, USA). The Student’s *t*-test or One-Way ANOVA was used to determine the significance value. *p*-value < 0.05 was considered significant. 

## Figures and Tables

**Figure 1 ijms-23-00887-f001:**
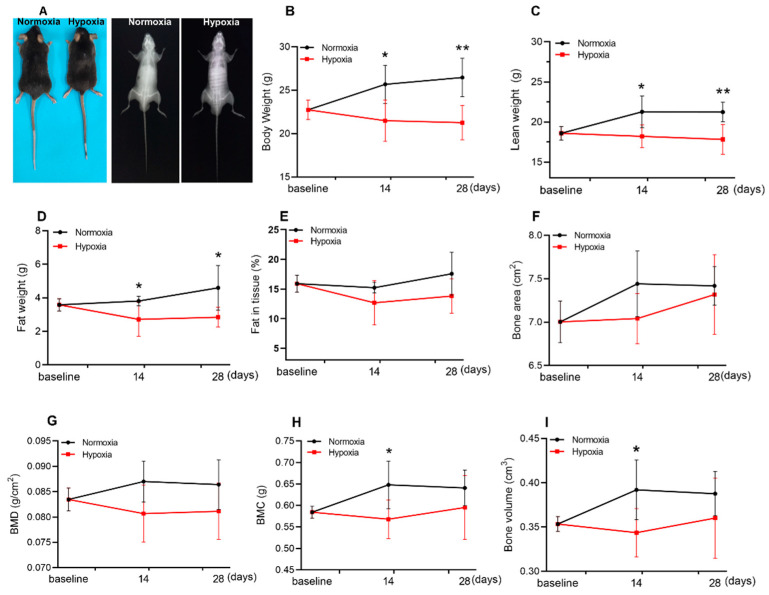
Hypoxia reduced the lean weight and fat weight of mice. Representative morphology (left) and dual X-ray digital images (right) of mice after 4 w of hypoxic treatment (**A**). DEXA results showed that body weight (**B**), lean weight (**C**), and fat weight (**D**) were significantly reduced after 14 d and 28 d of hypoxia. Fat in tissue (**E**), bone area (**F**), and BMD (bone mineral density) (**G**) were not affected in the whole experiment, while BMC (Bone mineral content) and (**H**), bone volume (**I**) decreased only at 14 d of hypoxia, and returned to normal levels at 28 d. Values represented means ± SD. * *p* < 0.05, ** *p* < 0.01. *n* = 5–6.

**Figure 2 ijms-23-00887-f002:**
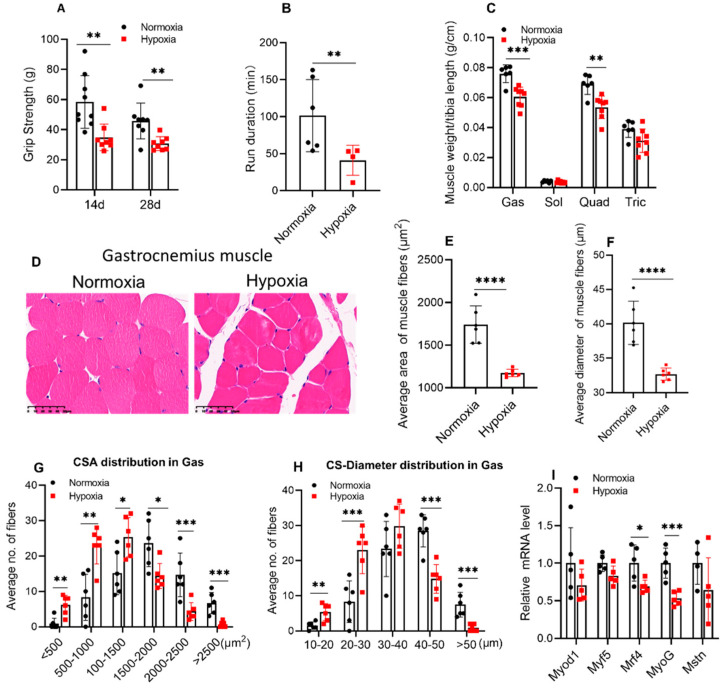
Four weeks of hypoxic exposure induced muscle atrophy of mice. The grip strength (**A**) of mice (*n* ≥ 8) and run duration time (**B**) in the treadmill experiment were significantly reduced after 4 w of hypoxia (*n* = 4–6). The weight of gastrocnemius (Gas) and quadriceps (Quad) muscles were reduced but not soleus (Sol) and triceps (Tric), of which were all normalized to the tibia length (**C**, *n* ≥ 6). (**D**) shows the representative images of H&E staining of Gas muscle. Scale bar, 50 μm. (*n* = 6). Hypoxia significantly decreased the average CSA (the cross-sectional area from the mid-belly of Gas muscle) and diameter (Feret’s diameter) in Gas muscles (**E**,**F**) and affected the area and diameter distribution of the myofibers (**G**,**H**, *n* = 6). The expression of myogenic factors *Mrf4* and *MyoG* significantly reduced in Gas muscles after 4 w of hypoxia (**I**, *n* = 5). Values represented means ± SD. * *p* < 0.05, ** *p* < 0.01, *** *p* < 0.001, **** *p* < 0.0001.

**Figure 3 ijms-23-00887-f003:**
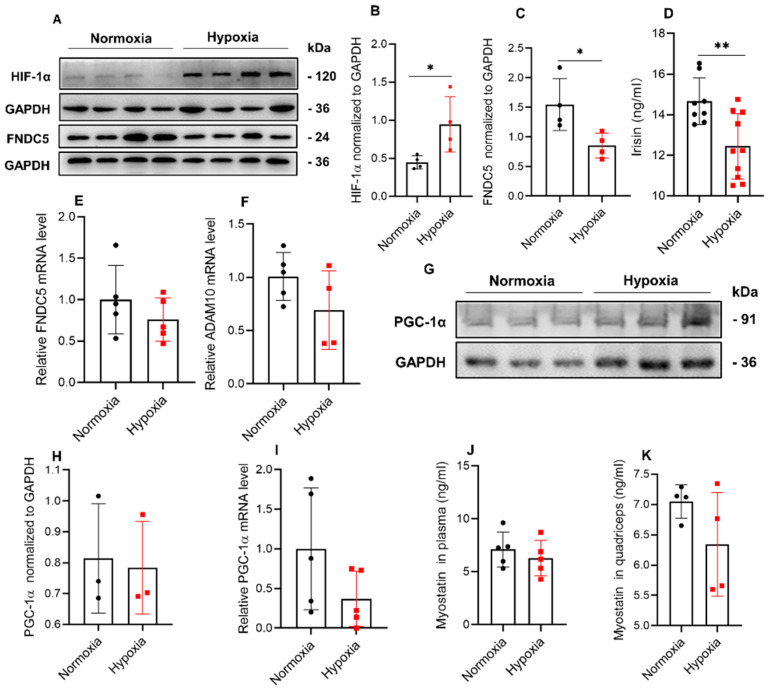
Hypoxia reduced the expression of FNDC5/irisin in mice. Hypoxic treatment significantly increased HIF-1α expression (**A**,**B**) and reduced FNDC5 expression (**C**) in Gas muscles (*n* = 4). Irisin concentration in plasma (**D**) of mice was reduced in hypoxia (*n* = 8–10), and the mRNA level of *FNDC5* (**E**) and *ADAM10* (**F**) in Gas muscle were not changed (*n* = 5). PGC-1α expression was not changed both in protein (**G**,**H**, *n* = 3) and mRNA level (**I**, *n* = 5). Myostatin concentration in both plasma (**J**, *n* = 5) and quadriceps muscle (**K**, *n* = 4) was not affected by hypoxia. Values represented means ± SD. * *p* < 0.05, ** *p* < 0.01.

**Figure 4 ijms-23-00887-f004:**
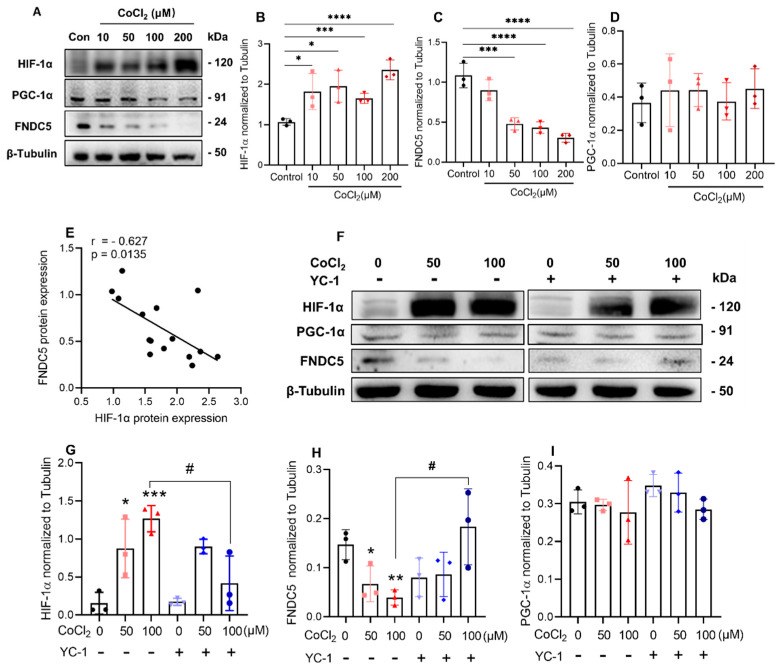
Inhibition of HIF-1α induced by CoCl_2_ increased FNDC5 expression in C2C12 myotubes (**A**), CoCl_2_ treatment all significantly increased the expression of HIF-1α (**B**,**C**), meanwhile decreased FNDC5 expression (**D**) and did not affect PGC-1α expression (**E**, *n* = 3). The expression of FNDC5 negatively correlated with HIF-1α, r > −0.5 represented the negative relationship (**E**). 50 μM of YC-1 treatment abrogated the increase in HIF-1α (**F**) and the decrease in FNDC5 (**G**) induced by CoCl_2_ (*n* = 3). PGC-1α expression was not affected by both CoCl_2_ (**H**) and YC-1 (**I**). Values represented means ± SD. * *p* < 0.05, ** *p* < 0.01, *** *p* < 0.001, **** *p* < 0.0001, ^#^ *p* < 0.05.

**Figure 5 ijms-23-00887-f005:**
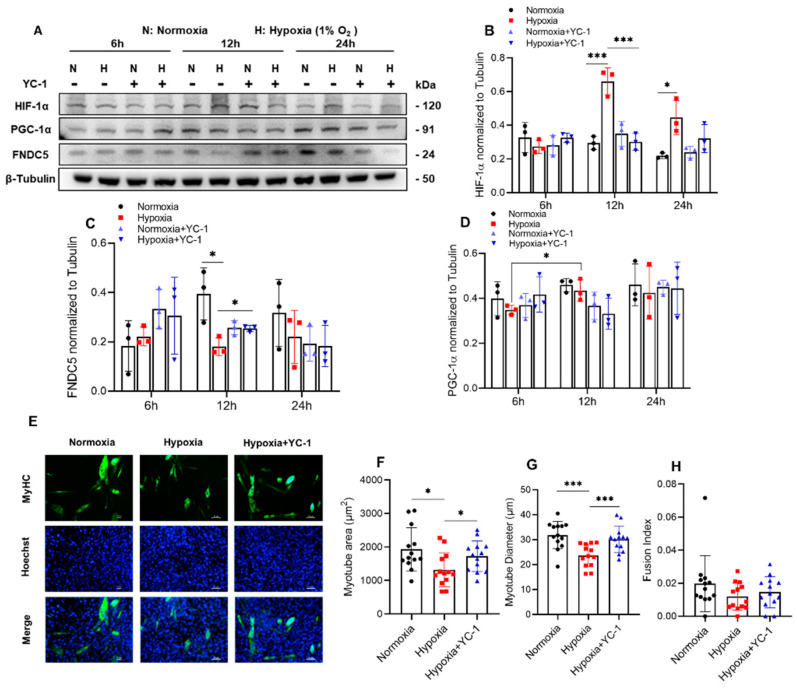
Inhibition of HIF-1α in 1% O_2_ ambient hypoxia increased FNDC5 expression, and myotube formation in C2C12 myotubes HIF-1α was increased significantly in hypoxia for 12h (**A**), which was rescued by YC-1 (**B**); meanwhile, YC-1 reversed the decrease in FNDC5 (**C**, *n* = 3). The expression of PGC-1α was not affected by both hypoxia and YC-1 (**D**). Hypoxia significantly reduced the area and diameter of C2C12 myotubes, and which were rescued by YC-1 treatment (**E**–**G**), while myotube fusion index was not influenced (**H**, *n* = 13). Values represented means ± SD. * *p* < 0.05, *** *p* < 0.001. Green, MyHC; blue, Hoechst. Scale bar, 50 µm.

**Table 1 ijms-23-00887-t001:** Primer sequences.

Primers	Forward Primer Sequences	Reverse Primer Sequences
*PGC-1α*	5′-CAACAATGAGCCTGCGAACA-3′	5′-CTTCATCCACGGGGAGACTG-3′
*FNDC5*	5′-GGACCTGGAGGAGGACACAGAATA-3′	5′-CTGGCGGCAGAAGAGAGCTATAA-3′
*Mstn*	5′-GGATGGCAAGCCCAAATGTT-3′	5′-GATTCAGGCTGTTTGAGCCA-3′
*GAPDH*	5′-CGGTGCTGAGTATGTCGTGG-3′	5′-ATGAGCCCTTCCACAATGCC-3′
*ADAM10*	5′-GGAAGCTTTAGTCATGGGTCTG-3′	5′-CTCCTTCCTCTACTCCAGTCAT-3′
*Myod1*	5′-TACGACACCGCCTACTACAGTG-3′	5′-GTGGTGCATCTGCCAAAAG-3′
*Myf5*	5′-CTGTCTGGTCCCGAAAGAAC -3′	5′-TGGAGAGAGGGAAGCTGTGT -3′
*MyoG*	5′-GCAATGCACTGGAGTTCG-3′	5′-ACGATGGACGTAAGGGAGTG-3′
*Mrf4*	5′-TGCTAAGGAAGGAGGAGCAA-3′	5′-CCTGCTGGGTGAAGAATGTT-3′

## Data Availability

The datasets generated in the current study are available from the corresponding author on reasonable request.

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
