# Peer review of "HIF-1α Negatively Regulates Irisin Expression Which Involves in Muscle Atrophy Induced by Hypoxia"

_ijms, 2022, doi:10.3390/ijms23020887_

Round 1

Reviewer 1 Report

Liu et al. Investigated the relationship between hypoxia induced atrophy, HIF-1a expression and irisin expression. They show that hypoxia induces muscle atrophy in mice and that both FNDC5 in muscle and irisin in serum are downregulated by hypoxia. In addition, they find the same effects in C2C12 myotubes upon CoCl2 treatment as well as hypoxic treatment. Importantly, they also show that HIF-1a inhibition rescues the hypoxia induced FNDC5 downregulation. They therefore conclude that hypoxia induces muscle atrophy via HIF-1a mediated FNDC5 downregulation and subsequent irisin reduction. The manuscript contains extensive in vivo and in vitro experiments and the findings are indeed interesting.

However, their rationale needs to be better described in the manuscript and the figures needs to be improved. In detail:

  • Irisin, FNDC5, ADAM10, PGC-1a and their relation to each other (i.e. PGC-1a can induce transcription of FNDC5, a transmembrane protein which is cleaved by ADAM10 to release irisin (type III domain of FNDC5)…..) should be described in the introduction. In addition, the role of the myogenic regulatory factors and their link to muscle atrophy or muscle physiology needs to be described. In this way, the readers will more easily follow the reason behind the different experiments and the authors should keep in mind that readers may not be familiar with muscle physiology and molecular and cellular mechanisms of muscle cells.
  • As the experiments with the HIF-1 inhibitor YC-1 belongs to the key experiments showing the link between HIF-1a and FNDC5, these experiments should be confirmed by using for instance HIF-1a siRNA.
  • Figure 2G and 2H: I have difficulties understanding these figures. What do the authors want to show here? There are no explanations in the text or figure legend. Why are the no. of fibers with small cross-sectional area increased by hypoxia and the no. of fibers with high cross-sectional area reduced upon hypoxia? And which fibers are shown in E, F and I,J? And why do the authors show the quantification of fibers >50µm in H and the fibers <500µm in G when they are all shown in each graph? Please add better explanations in the figure legend for these graphs.
  • Figure 1: Please add a caption for the x-axis (days)
  • Figure 3: Do the authors have another better blot for A? it does not mirror the quantitative results in C very well, as the FND5C only seems to be expressed stronger in lane 3 and 4 and no difference to lane 7. I would also recommend to use a different housekeeping protein than GAPDH, as GAPDH is influenced by hypoxia. K) Why did the authors detect Myostatin and not Mrf4 or Myogenin, which were downregulated in figure 2M?
  • Figure 4: Why is HIF-1a expression levels lower under 100µM coCl2? I) The blot is not convincing, do the authors have a better one?
  • Figure 5: a) Please add the denotation for N and H in the figure legend. B) the colours for the legend are missing; which bar is which treatment? C) the legend is missing, which bar is which treatment? Please include this in both figures. Also, the difference between hypoxia and hypoxia + YC-1 in FND5C protein band density at 12h cannot be significantly different as noted in the graph (if I interpreted the bars correctly, which was not easy as the legend is missing). To me, it looks as if YC-1 treatment did not rescue FNDC5 expression. E) the immunofluorescent images with YC-1 treatment are not convincing in my opinion. Maybe the authors have a better image, where the difference to sole hypoxia treatment is more visible?
  • In overall, the figures have a very bad resolution and it is difficult to separate significance stars from data points, scale bars are hardly visible and captions for y-axis and x-axis are sometimes not readable. Please provide high resolution images and place them at a larger size in the manuscript.
  • For many experiments, the number of experiments (n) is missing in the figure legends. Please insert this everywhere, where it is missing.
  • Materials and methods: Please add the % O2 to the altitude of 399m in section 4.1. Please describe how blood samples from mice were drawn (4.4). Please add the source of YC-1 (4.7). Please add the source of the C2C12 myoblasts (4.6).
  • General: Please explain abbreviations the first time they are used (for instance DEXA, CSA, FNDC5 etc.)
  • The manuscript needs to be checked by a native speaker, as many sentences need to be rephrased and are grammatically wrong.

Reviewer 2 Report

Ref: ijms-1498244

Title: HIF-1α negatively regulates irisin expression which involves in muscle atrophy induced by hypoxia

MAJOR REVISION

The aim of the study was to investigate the interplay between HIF-1α and irisin in muscle atrophy caused by hypoxia in in vivo and in vitro models. The main result is that the Authors identified that irisin is negatively regulated by HIF-1α. The conducted research is interesting. The main disadvantage of the study is that the Authors did not use the treatment – HIF1a inhibitor – lificiguat (YC-1) in the in vivo model. The Authors verified their hypotheses with the HIF1a inhibitor only in in vitro model.  

Additional comments:

  1. Why the Authors use 2 different reference protein during western blot i.e., b-tubulin and GAPDH.
  2. The Authors measure many parameters, many mRNAs/proteins but there is no explanation in the text why they chose these markers e.g., PGC1a.
  3. Majority of the Figures is too small to be readable e.g. Fig. 2, Fig. 5.
  4. I do not understand why the Authors differentially presents the qPCR results. In the Fig. 2 – normoxia is the 1-fold whereas in the e.g., Fig. 3 E, I the normoxia is not 1-fold. Please, present qPCR control results as 1-fold.
  5. Please avoid the abbreviation in the Discussion part e.g., Gas, Sol.
  6. Why the Authors use only male mice?
  7. How the Authors chose the reference gene in the qPCR method?
  8. The Authors use high concentrations of anti-b-tubulin and anti-GAPDH. It is too high to see the differences in sample input.
  9. The density of the cells should be added.

Round 2

Reviewer 1 Report

The manuscript of Liu et al. has improved. Introduction and figures are more clear now. However, I still have some minor points regarding the language, which should be considered:

  • Lines 46-49: Please rephrase these sentences in the present tense (i.e. use “is” instead of “was”)
  • Lines 51-54: these sentences are too long and should be rephrased and separated. For instance, start a new sentence with “In vitro” in line 52.

Figure 2: Please add the explanation for CSA in the figure legend.

Reviewer 2 Report

The Authors addressed all my concerns / questions. 

In my opinion, the manuscript is ready for publication.

Author Response

Thanks very much for your kind work and valuable comments.